# Hemoglobin Levels, Anemia, and Their Associations with Body Mass Index among Pregnant Women in Hail Maternity Hospital, Saudi Arabia: A Cross-Sectional Study

**DOI:** 10.3390/nu15163508

**Published:** 2023-08-09

**Authors:** Reem Eltayeb, Naif K. Binsaleh, Ghaida Alsaif, Reem M. Ali, Amjad R. Alyahyawi, Ishag Adam

**Affiliations:** 1Department of Medical Laboratory Science, College of Applied Medical Science, University of Ha’il, Ha’il 8227, Saudi Arabia; n.binsaleh@uoh.edu.sa (N.K.B.); g.alsaif@uoh.edu.sa (G.A.); re.ali@uoh.edu.sa (R.M.A.); 2Department of Diagnostic Radiology, College of Applied Medical Sciences, University of Ha’il, Ha’il 8227, Saudi Arabia; a.alyahyawi@uoh.edu.sa; 3Centre for Nuclear and Radiation Physics, Department of Physics, University of Surrey, Guildford GU2 7XH, UK; 4Department of Obstetrics and Gynecology, Unaizah College of Medicine and Medical Sciences, Qassim University, Unaizah 51911, Saudi Arabia; ia.ahmed@qu.edu.sa

**Keywords:** pregnancy, anemia, body mass index

## Abstract

The level of association between hemoglobin level/anemia and obesity during pregnancy is not yet fully understood. A cross-sectional study was conducted in Hail Maternity Hospital in northern Saudi Arabia from March to May 2023 to assess the associations between hemoglobin levels, anemia, and obesity among pregnant women. Reproductive history was gathered through a questionnaire. Body mass index (BMI) was calculated/computed from the women’s weight and height. Linear and binary regression analyses were performed. A total of 334 pregnant women were enrolled. The median (interquartile range (IQR)) age and parity were 33.2 (27.4–37.8) years and 3 (1–5), respectively. Of the 334 women, 52 (15.6%) were normal weight, while 87 (26.0%) were overweight and 195 (58.4%) were obese. In the multiple linear regression, parity (coefficient = −0.121, *p* = 0.001) and BMI (coefficient = 0.030, *p* = 0.006) were negatively associated with the hemoglobin level. Ninety-nine (26.9%) women had anemia. In the multivariate analysis, primiparity (adjusted odds ratio (AOR) = 0.54, 95% confidence interval (CI) = 0.30–0.97), increasing BMI (AOR = 0.93, 95% CI = 0.89–0.97), and obesity (AOR = 0.31, 95% CI = 0.16–0.61) were associated with decreased ORs of anemia. Increasing parity was associated with increased ORs of anemia (AOR = 1.18, 95% CI = 1.05–1.32). Being overweight was not associated with anemia (AOR = 0.56, 95% CI = 0.27–1.16). In the current study, a high hemoglobin level and lower prevalence of anemia were observed among obese pregnant women.

## 1. Introduction

Anemia during pregnancy, especially among women in countries with less resources, is one of the main health problems that can increase the risk for both maternal and perinatal adverse outcomes [1]. The “World Health Organization (WHO)” considered/defines anemia among pregnant women as a hemoglobin level of less than 11 g/dL and anemia is considered severe when the hemoglobin level during pregnancy is less than 7 g/L [2]. Moreover, if the prevalence of anemia in a population is ≥40%, anemia is considered a severe public health risk in the index population [3]. Although there are many causes of anemia during pregnancy, iron deficiency is considered the most common cause of anemia among pregnant women [2]. Anemia during pregnancy is one of the main worldwide health problems; however, certain regions in the world have a high prevalence of anemia during pregnancy; for example, while 24.1% of pregnant women in South America have anemia, over half (57.0%) of pregnant women in African countries and near half (48%) of pregnant women in South-East Asia have anemia [3]. Moreover, a recent meta-analysis that included worldwide studies reported that the pooled prevalence of anemia during pregnancy is 36.8% [4]. Several factors, such as infections (malaria), poor utilization of health services (antenatal care (ANC)), residency, a low level of education, a short interpregnancy interval, and non-taking supplements in the form of iron-folic acid, are associated with increased risk of anemia during pregnancy [5,6]. Previous studies have shown that anemia during pregnancy, especially its severe form, is one of the main predictors for several maternal and perinatal poor outcomes, such as increased risk of cesarean delivery, preterm birth, low birthweight, and both maternal and perinatal mortality [7,8,9,10,11]. Therefore, the WHO recommends hemoglobin testing during pregnancy (each trimester) as a tool for screening for iron deficiency anemia [12]. However, iron deficiency during pregnancy is considered when serum ferritin is less than <15 μg/L (in the absence of inflammation) and iron deficiency anemia is defined as hemoglobin less than 11 g/dL and serum ferritin less than <15 μg/L [13].

There are increasing rates of obesity among women of reproductive age especially in in middle- and low-income countries and the factors responsible for the increasing rates are multifactorial, which include genetic, environmental, physiologic, socioeconomic, and cultural factors [14]. Obesity during pregnancy is one of the main worldwide health problems, and it can lead to several adverse maternal and perinatal outcomes, such as gestational diabetes mellitus, hypertensive disorders of pregnancy, labor induction, chorioamnionitis, and macrosomia [15,16]. Several factors such as age and educational level are associated with obesity during pregnancy [17]. Previous studies have shown different levels of association between BMI and obesity and anemia or hemoglobin levels among pregnant women [16,18,19,20,21,22,23,24,25,26,27,28,29]. While some studies have shown that hemoglobin levels were significantly higher among pregnant women with obesity [18,27] or that pregnant women with obesity were less likely to have anemia [22,23,28], other studies have shown that hemoglobin levels were significantly lower in women with obesity [29] and that obese pregnant women were at higher risk for anemia [16,24]. Some studies have shown no association between BMI and anemia among pregnant women [19,20,21], while other studies have shown that a lower BMI was associated with anemia [25,26].

Previous studies have shown a high prevalence of obesity and overweight among Saudi women and found that obese pregnant women were at higher risk for gestational diabetes mellitus, labor induction, hypertension, operative delivery, and macrosomia [15,16]. Moreover, a recent study showed that 34.1% of pregnant women in central Saudi Arabia had anemia [23]. Therefore, the association between anemia and obesity during pregnancy needs to be explored to have evidence-based data necessary for preventive measures. The current study was conducted to assess hemoglobin levels, anemia, and their associations with BMI and obesity among pregnant women in Hail Hospital, Saudi Arabia.

## 2. Materials and Methods

### 2.1. Study Setting and Design

A cross-sectional study was conducted in the antenatal care department of Hail Maternity Hospital in northern Saudi Arabia from March to May 2023.

#### 2.1.1. Inclusion Criteria

Adult pregnant women (aged ≥18 years) with singleton pregnancies in their early pregnancy (≤14 weeks of gestational age) presenting to the antenatal care department of Hail Hospital were enrolled in this study after signing an informed written consent form.

#### 2.1.2. Exclusion Criteria

Pregnant women w > 14 weeks of gestational age or who had multiple pregnancies, vaginal bleeding in the current pregnancy, or systemic diseases such as diabetes mellitus, hypertension, thyroid disease, hemolytic disease, or intrauterine fetal demise were excluded.

Systematic random sampling was used to select pregnant women who fulfilled the inclusion criteria. In the previous hospital records, 648 pregnant women in their early pregnancies attended the antenatal care department over the three-month period prior to the study. The sampling interval (≈2) was detected by dividing the expected number of pregnant women (648) in the study period by the required sample size, which was 334 women (648/334). The pregnant women were therefore interviewed at three intervals until the required sample size (334) was reached.

After the pregnant women signed their informed consent forms, two female medical officers interviewed all the participants enrolled in the study using a questionnaire on the women’s demographic, clinical, and reproductive health information. The information collected was age, number of deliveries (parity), education level, history of previous miscarriage, number of ANC visits, and intake of folic acid (iron is not recommended during the first trimester of pregnancy). Following standard procedures, the women’s weight and height were then measured to calculate their BMI, which was grouped according to the WHO classification: underweight (<18.5 kg/m^2^), normal weight (18.5–24.9 kg/m^2^), overweight (25.0–29.9 kg/m^2^), or obese (≥30.0 kg/m^2^) [30]. Then, each woman had 2 mL of blood drawn into an ethylenediaminetetraacetic acid tube. These samples were used to analyze the hemoglobin level, which was performed according to the manufacturer’s instructions (Sysmex KX-21, Kobe, Japan).

### 2.2. Sample Size

The sample size (N) of 334 pregnant women was estimated with an assumed prevalence of anemia of 30% (1:2 ratio between anemic and non-anemic women), which has previously been reported in the study area (Hail) of Saudi Arabia [23], using the single proportional formula n = Z^2^pq/d^2^, where q = (1 − p), Z1 − α (95% confidence interval (CI)) = 1.96, and d (margin of error of 5%) = 0.05. We assumed that 25.0% of anemic women would have obesity, while 40.0% of non-anemic women would have obesity. OpenEpi Menu was used to calculate the sample size [31].

### 2.3. Statistics

SPSS for Windows version 22 (IBM, Armonk, NY, USA) was used to analyze the data. The continuous data (age, parity, BMI, and hemoglobin) were checked for normality using the Shapiro–Wilk test, and the data were not normally distributed. Medians (interquartile ranges (IQRs)), frequencies, and proportions were used to describe the women’s characteristics. Chi-square and non-parametric tests were used to compare the variables between the women with anemia and those without anemia, as appropriate. Univariate analyses were performed with anemia (binary) and hemoglobin as the dependent variables. The included independent variables were age, parity, previous history of miscarriage, BMI, and folic acid intake. After evaluating the presence of multicollinearity (if the variance inflation factor exceeded 4), variables with a *p*-value < 0.2 were used to build both the multivariable and multilinear analysis to evaluate and control the effects of each covariate on the other variables. The results of the adjusted odds ratios (AORs) with 95% CIs, coefficients, and standard errors were considered final, and a *p*-value less than 0.05 was considered significant.

## 3. Results

A total of 334 pregnant women were enrolled. The median (IQR) age and parity were 33.2 (27.4–37.8) years and 3 (1–5), respectively. Of the 334 women, 135 (40.4%) were educated to secondary school level or higher, and 199 (59.6%) were educated to less than secondary school level. Of these 334 women, 124 (37.1%) had a history of miscarriage and 254 (76.0%) had fewer than two ANC visits. A total of 138 women had a history of taking folic acid. The median (IQR) BMI of the women was 30.4 (26.8–33.7) kg/m^2^. Of the 334 women, while 52 (15.6%) were normal weight, 87 (26.0%) were overweight and 195 (58.4%) were obese (Table 1). No woman was underweight.

Comparing with the pregnant women with normal weight, the women with obesity had significantly higher ages, parity, and hemoglobin levels (Figure 1).

Education school level, level of ANC visits, history of miscarriage, and taking folic acid were not different between the women with obesity and those with normal weight. Age, parity, and hemoglobin levels were not different between the women with overweight and those with obesity (Table 2).

The median (IQR) hemoglobin level was 11.8 (10.6–12.7) g/dL. In the multiple linear regression, while parity (coefficient = −0.121, *p* = 0.001) and BMI (coefficient = 0.030, *p* = 0.006) were associated with hemoglobin level, age, education level, level of ANC visits, a history of miscarriage, and taking iron-folic acid were not (Table 3).

Ninety-nine (26.9%) women had anemia, and one (0.3%) woman had severe anemia (hemoglobin < 7/dL). In the univariate analysis, parity, BMI, and obesity were associated with OR (unadjusted) of anemia. Age, being a primipara, education level, level of ANC visits, a history of miscarriage, and taking folic acid were not associated with OR (unadjusted) of anemia. Compared with the non-anemic women, a significantly lower number of anemic women were obese (46/99 (46.9%) vs. 149/235 (63.4%)) (Table 4).

In the multivariate analysis, primiparity (adjusted odds ratios, OR = 0.54, 95% CI = 0.30–0.97), increasing BMI (AOR = 0.93, 95% CI = 0.89–0.97), and obesity (AOR = 0.31, 95% CI = 0.16–0.61) were associated with a lower risk of anemia. Increasing parity was associated with a higher risk of anemia (AOR = 1.18, 95% CI = 1.05–1.32). Being overweight was not associated with anemia (AOR = 0.56, 95% CI = 0.27–1.16) (Table 5).

## 4. Discussion

The main result of the current study was that the pregnant women with obesity had a higher hemoglobin level and a lower risk for anemia. This is in agreement with several previous studies [18,23,28,32]. In Saudi Arabia, BMI in pregnant women has been reported to be negatively associated with anemia, and obesity reduced the risk of anemia by 10.0% (AOR = 0.90) [23]. We previously reported that hemoglobin levels were significantly higher in obese pregnant women (*n* = 101) compared with women with normal weight (*n* = 95) among Sudanese women in early pregnancy [18]. Moreover, Mocking et al. observed that hemoglobin levels were higher in early pregnancy in pregnant women with obesity in Ghana and Indonesia, and the risk for anemia decreased with a higher early pregnancy BMI [32]. Likewise, Liabsuetrakul et al. reported that women with overweight and obesity (pre-pregnancy) had a lower prevalence of anemia [28]. Their reported BMI values were higher than the values in “international criteria-based BMI” (22.4% and 10.1% vs. 15.5% and 3.4%, respectively). In a large national cross-sectional study conducted in 16 provinces in six regions in China, of 11,782 pregnant women, the women with overweight and obesity had a lower risk of iron deficiency anemia (AOR = 0.68 and AOR = 0.30, respectively) [33]. In a population-based cross-sectional study that enrolled 5,679,782 women in China using pre-pregnancy BMI, the women with overweight and obesity were found to be at lower risk of anemia (AOR = 0.84 and AOR = 0.70, respectively) [34]. In another study that enrolled 1546 pregnant women in Peking University International Hospital, Beijing, China, pre-pregnancy obesity was associated with a lower risk of anemia (OR = 0.30) [35].

In India, a large cohort study of pregnant women showed that the prevalence of anemia was significantly higher (32.1%) in the women with underweight compared with the prevalence of anemia in women with normal weight (4.2%) and with obesity (3.9%) [26]. Likewise, in Nepal, in 609 pregnant women, anemia was associated with a low BMI (AOR) [25].

On the other hand, in Morocco, anemia was more prevalent (58.8%) among pregnant women with obesity (*n* = 390) [24]. One study reported that pregnant women with obesity (*n* = 23) in their second trimester had a significantly lower hemoglobin level (10.8 g/dL) compared with the hemoglobin level (11.5 g/dL) of an equal number of pregnant women of normal weight [29].

In the Nigeria Demographic and Health Survey, Ezenweke et al. [36] reported a non-linear (cubic) effect of BMI on hemoglobin levels. Their results indicated that the hemoglobin levels of pregnant women decreased with BMI until the lower limit or cutoff point at a BMI value of about 23 kg/m^2^. Thereafter, a steady increase was reported until a BMI cutoff point of 34 kg/m^2^ [36]. Previous studies showed no association between BMI and anemia among pregnant women in central Sudan [19] and in Khartoum, Sudan [20,21]. The different results of these studies could be explained by different sociodemographic characteristics, differences in the prevalence of anemia and obesity, and different inclusion and exclusion criteria.

In the current study, parity was significantly higher among the women with obesity, and increasing parity was associated with a higher risk of anemia (AOR = 1.18). Previous studies have shown that increasing parity in women is associated with an increased risk of obesity [37,38], and parity is associated with iron deficiency [39]. Perhaps repeated pregnancies depleted body stores of iron and resulted in iron deficiency [40]. The exact explanations for the association between obesity and anemia have not been fully explored, although a previous study has shown that obesity is positively associated with serum ferritin [41]. Obesity may influence iron homeostasis due to its effect on hepcidin levels, which are mediated by chronic inflammation [42]. Other factors that could explain the association between obesity and anemia are physical activity and the quality of nutrition [42,43]. The limitations of our research were that the nature of this cross-sectional study could not identify the direction of association; a longitudinal study could be more helpful in exploring the association between BMI and anemia. The types of anemia were not identified, and several factors such as serum ferritin, vitamin D, hepcidin, and other inflammatory factors were not assessed.

## 5. Conclusions

The current study found higher blood hemoglobin levels and lower anemia prevalence among pregnant women with obesity.

## Figures and Tables

**Figure 1 nutrients-15-03508-f001:**
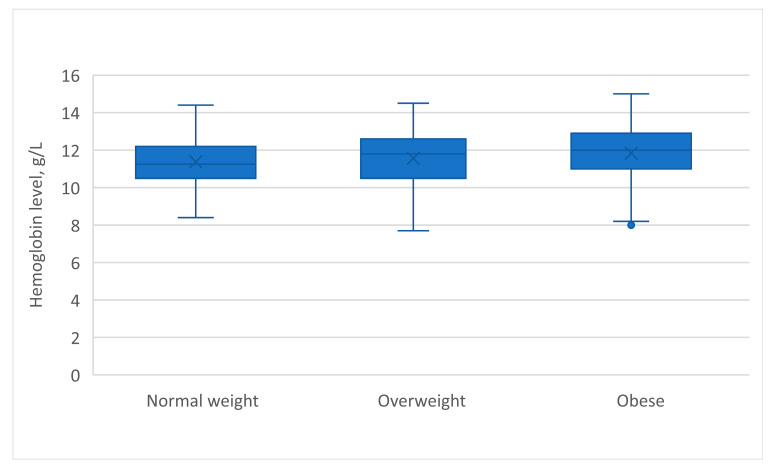
Comparing hemoglobin level between normal weight, overweight, and obese pregnant women.

**Table 1 nutrients-15-03508-t001:** General sociodemographic and the clinical characteristics of pregnant women (N = 334) in Hail, Saudi Arabia, 2023.

Variable		*Median*	*Interquartile Range*
Age, years		33.2	27.4–37.8
Parity		3	1–5
BMI, kg/m^2^		30.4	26.8–33.7
Hemoglobin level, g/dL		11.8	10.6–12.7
		** *Frequency* **	** *Proportion* **
Parity	Primipara	88	26.3
	Parous	246	73.7
Education school level	≥Secondary	135	40.4
	<Secondary	199	59.6
History of miscarriage	No	210	62.9
	Yes	124	37.1
Antenatal care visits	>2	80	24.0
	≤2	254	76.0
Taking folic acid	Yes	138	41.3
	No	196	58.7
Body mass index, kg/m^2^	Normal weight	52	15.6
	Overweight	87	26.0
	Obese	199	58.4

**Table 2 nutrients-15-03508-t002:** Sociodemographic and clinical characteristics of women in Hail, Saudi Arabia, by BMI.

Variable		Normal Weight (*n* = 52)	Overweight (*n* = 87)	Obese (*n* = 195)	*p*
*Median (interquartile range)*					
Age, years		27.8 (24.5–32.4)	34.5 (29.0–37.9)	33.8 (28.6–38.7)	Normal weight vs. overweight, *p* < 0.001 Normal weight vs. obese, *p* < 0.001 Overweight vs. obese, *p* = 0.496
Parity		2 (1–3)	3 (2–4)	3 (2–5)	Normal weight vs. overweight, *p* = 0.009 Normal weight vs. obese, *p* = 0.001 Overweight vs. obese, *p* = 0.853
Hemoglobin level, g/L		11.2 (10.5–12.2)	11.8 (10.5–12.6)	12.0 (11.0–12.9)	Normal weight vs. overweight, *p* = 0.875 Normal weight vs. obese, *p* = 0.015 Overweight vs. obese, *p* = 0.339
*Frequency (proportion)*					
Parity	Primipara	22 (42.3)	19 (21.8)	47 (24.1)	0.016
	Parous	30 (57.7)	68 (78.2)	148 (75.9)	
History of miscarriage	No	38 (73.1)	59 (67.8)	113 (57.9)	0.072
	Yes	14 (26.9)	28 (32.2)	82 (42.1)	
Education school level	≥Secondary	22 (42.3)	38 (43.7)	75 (38.5)	0.680
	<Secondary	30 (57.7)	49 (56.3)	120 (61.5)	
Antenatal care visits	>2	11 (21.2)	20 (23.0)	49 (25.1)	0.812
	≤2	41 (78.8)	67 (77.0)	146 (74.9)	
Taking folic acid	Yes	17 (32.7)	42 (48.3)	79 (40.5)	0.184
	No	35 (67.3)	45 (51.7)	116 (59.5)	

**Table 3 nutrients-15-03508-t003:** Multiple linear regression of factors associated with hemoglobin levels among pregnant women in Hail, Saudi Arabia, 2023.

		Coefficient	Standard Error	95% CI	*p*
Age, years		0.004	0.006	−0.007–0.015	0.430
Parity		−0.121	0.037	−0.194–−0.049	0.001
BMI, kg/m^2^		0.030	0.011	0.009–0.052	0.006
History of miscarriage	No	Reference			0.339
	Yes	0.156	0.163	−0.164–0.476	
Education school level	≥Secondary	Reference			0.565
	<Secondary	0.360	0.630	−0.087–0.160	
Antenatal care visits	>2	Reference			0.091
	≤2	0.334	0.197	−0.054–0.722	
Taking folic acid	Yes	Reference			0.070
	No	0.171	−0.094	0.006–0.009	

**Table 4 nutrients-15-03508-t004:** Univariate analysis (unadjusted) of factors associated with anemia in Hail, Saudi Arabia, 2023.

Variable		Women with Anemia (*n* = 99)	Women without Anemia (*n* = 235)	Odds Ratio (95% CI)	*p*
*Median (interquartile range)*					
Age, years		33.7 (29.2–38.3)	32.7 (27.1–37.6)	0.99 (0.98–1.01)	0.831
Parity		4 (2–5)	3 (1–4)	1.12 (1.01–1.25)	0.024
BMI, kg/m^2^		29.3 (25.2–32.8)	31.6 (27.5–36.2)	0.94 (0.90–0.98)	0.004
*Frequency (proportion)*					
Parity	Primipara	20 (20.2)	68 (28.9)	0.62 (0.35–1.09)	0.100
	Parous	79 (79.8)	167 (71.1)	Reference	
History of miscarriage	No	66 (66.7)	144 (61.3)	Reference	0.352
	Yes	33 (33.3)	91 (38.7)	0.79 (0.48–1.29)	
Education school level	≥Secondary	56 (56.6)	79 (33.6)	Reference	
	<Secondary	43 (43.4)	156 (66.4)		
Antenatal care level	>2 visits	23 (23.2)	57 (24.3)	Reference	0.814
	≤2 visits	76 (76.8)	178 (75.7)	0.94 (0.54–1.64)	
Taking folic acid	Yes	45 (45.5)	93 (39.6)	Reference	0.319
	No	54 (54.5)	142 (60.4)	1.27 (0.79–2.04)	
Body mass index, kg/m^2^	Normal	23 (23.2)	27 (11.5)	Reference	
	Overweight	30 (30.3)	57 (24.3)	0.66 (0.32–1.34)	0.253
	Obese	46 (46.5)	149 (63.4)	0.38 (0.20–0.73)	0.004

**Table 5 nutrients-15-03508-t005:** Adjusted multiple binary regression analysis of factors associated with anemia in Hail, Saudi Arabia.

Variable		Adjusted Odds Ratio (95% Confidence Interval)	*p*
Parity		1.18 (1.05–1.32)	0.003
Parity	Primipara	0.54 (0.30–0.97)	0.042
	Parous	Reference	
Body mass index, kg/m^2^		0.93 (0.89–0.97)	0.001
Body mass index, kg/m^2^	Normal	Reference	
	Overweight	0.56 (0.27–1.16)	0.122
	Obese	0.31 (0.16–0.61)	0.001

## Data Availability

The data presented in this study are available on request from the last author.

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
