# Peer review of "Hemoglobin Levels, Anemia, and Their Associations with Body Mass Index among Pregnant Women in Hail Maternity Hospital, Saudi Arabia: A Cross-Sectional Study"

_nutrients, 2023, doi:10.3390/nu15163508_

Round 1

Reviewer 1 Report

This is a good manuscript with beneficial information on the association of anemia with normal weight, overweight or obesity. Nonetheless, this MS can be improved with the following recommendations:

(1) The term "anemia" in this MS needs to be defined specifically. There are different types of anemia. I presume that the anemia you referred to in this MS is "iron defficiency anemia". Need confirmation.

(2) Provide clarity on the definition of anemia, iron deficiency anemia (IDA) and iron deficiency.

(3) IDA is defined in terms of Hemoglobin levels as indicated in Line 40; whereas ID is defined in terms of the level of ferritin. It is very confusing in this study when data values of hemoglobin which were used in correlation, univariate and multivariate regression analyses are used to describe anemia.

(4) Lines 103 -104 stated that complete blood count was analyzed in this study. It would benefit the readers if you can provide more information on the blood counts analytes including the hemoglobin concentration. Were other analytes such as ferritin, RBC, WBC, blood volume, etc analyzed?

(5) Line 186. What is the definition of "severe anemia" as referred to here?

(6) Provide possible factors why parity is positively correlated with anemia. Comment on whether bleeding during delivery and placenta expulsion could be a possible contributor to why muliparous births are associated with higher anemia.

MINOR EDITS are;

(1) Add "women" after "among pregnant" in Line19.

(2)  Add "school" between "secondary" and "level" in Lines 134-135, in Table 1 and other places in the MS.

(3) In Figure 1 title in Line 165, change "Comparing" to " Comparison of" before "hemoglobin level ........"

Needs a slight improvement.

Author Response

We would like to thank the editor and the reviewers for their valuable comments on the manuscript and we feel that their comments/response have improved the manuscript dramatically

Reviewer # 1

Comments and Suggestions for Authors

This is a good manuscript with beneficial information on the association of anemia with normal weight, overweight or obesity. Nonetheless, this MS can be improved with the following recommendations:

Response

We do appreciate your positive comments and thank you very much for your time.

Comment

(1) The term "anemia" in this MS needs to be defined specifically. There are different types of anemia. I presume that the anemia you referred to in this MS is "iron deficiency anemia". Need confirmation.

Response

Okay, we agreed and the definition of anemia, iron deficiency and iron deficiency anemia is defined through out the MS, Please see lines 40, 41, 44, 45, 59-62.

Comment

(2) Provide clarity on the definition of anemia, iron deficiency anemia (IDA) and iron deficiency.

Response

Yes, we agreed and these have been defined. Please see lines Please see lines 40, 41, 44, 45, 59-62

Comment

(3) IDA is defined in terms of Hemoglobin levels as indicated in Line 40; whereas ID is defined in terms of the level of ferritin. It is very confusing in this study when data values of hemoglobin which were used in correlation, univariate and multivariate regression analyses are used to describe anemia.

Response

Yes, I think, it is now obvious after the above definitions which are inserted in the MS. Thus, we measured hemoglobin only and we did not measured ferritin (Please see line 273). This mean that we are assessing anemia and from the title of the MS and throughout we mean anemia.

Comment

(4) Lines 103 -104 stated that complete blood count was analyzed in this study. It would benefit the readers if you can provide more information on the blood counts analytes including the hemoglobin concentration.  Were other analytes such as ferritin, RBC, WBC, blood volume, etc analyzed?

Response

Yes, we agreed. Unfortunately, we did not record the other factors, RBC, WBC, blood volume, etc analyzed?

So, we now edit the method and we mentioned hemoglobin only. Please see line 122.

Comment

(5) Line 186. What is the definition of "severe anemia" as referred to here?

Response

Yes it is defined. Please see lines 42, 196

Comment

(6) Provide possible factors why parity is positively correlated with anemia. Comment on whether bleeding during delivery and placenta expulsion could be a possible contributor to why muliparous births are associated with higher anemia.

Response

Yes, we agreed and we inserted the explanation. Please see lines 263-264

Comment

MINOR EDITS are;

(1) Add "women" after "among pregnant" in Line19.

   Response       

Yes, added. Please see line 20

Comment          

(2)  Add "school" between "secondary" and "level" in Lines 134-135, in Table 1 and other places in the MS.

Response

Yes, added as suggested.

Comment

(3) In Figure 1 title in Line 165, change "Comparing" to " Comparison of" before "hemoglobin level ........"

Response

Yes corrected

Comment

Comments on the Quality of English Language

Needs a slight improvement.

Response

Yes, improved

Regards

Reviewer 2 Report

Generally, you have presented an original and very interesting study, of clinical significance.

There are few remarks/corrections i would suggest to consider:

1) At the abstract section you refer the period: "January to March 2023", and at the materials section you refer the period: "March to May 2023", what is the correct period of your study selection?

2) At the Statistics section (page 3), you have to correct the word "dat" at the 1st line of this paragraph with the correct word "data".

3) At the Materials and Methods section, (page 3, 2nd paragraph from the above): Except the intake of folic acid from the pregnant women of your study did you ask how many women of your study did they take iron supplementation, during their pregnancy?

4) At the References section can you provide the relevant website for the reference No. 3? Do you mean WHO, or GHO?

Moreover can you provide more data and the relevant DOI number for the reference No. 7?

5) At the Results section can you analyze the OR= odds ratio and AOR=adjusted odds ratio at the text, in order to be more clear and better presented?

Congratulations for your study!!

Author Response

We would like to thank the editor and the reviewers for their valuable comments on the manuscript and we feel that their comments/response have improved the manuscript dramatically

Reviewers # 2

Generally, you have presented an original and very interesting study, of clinical significance.

Response

Thank you so much and we do appreciate your positive response

Comment

There are few remarks/corrections i would suggest to consider:

  • At the abstract section you refer the period: "January to March 2023", and at the materials section you refer the period: "March to May 2023", what is the correct period of your study selection?

Corrected ( it is from March to May 2023)

Response

Yes, corrected and deep apologize

Comment

  • At the Statistics section (page 3), you have to correct the word "dat" at the 1st line of this paragraph with the correct word "data".

Response

Corrected as suggested. Please see line 135

Comment

  • At the Materials and Methods section, (page 3, 2nd paragraph from the above): Except the intake of folic acid from the pregnant women of your study did you ask how many women of your study did they take iron supplementation, during their pregnancy?

Response

    Yes, it is valid point. We enrolled women in early pregnancy, iron is not recommended during early pregnancy (perhaps it will increase nausea?). This point has been inserted, please see line 116-117

Comment

4) At the References section can you provide the relevant website for the reference No. 3? Do you mean WHO, or GHO?

Global Health Observatory data repository by WHO

(Website provided )

Moreover can you provide more data and the relevant DOI number for the reference No. 7?

DOI number Provided

Response

Yes, references are checked thoroughly and these points corrected.

Comment

5) At the Results section can you analyze the OR= odds ratio and AOR=adjusted odds ratio at the text, in order to be clearer and better presented?

Response

Yes, it has been edited. Please see line 197-204

Regards